# Analysis of KSHV B lymphocyte lineage tropism in human tonsil reveals efficient infection of CD138+ plasma cells

**Farizeh Aalam**◯ᵒ, **Romina Nabiee**◯ᵒ, **Jesus Ramirez Castano**◯, **Jennifer Totonchy**◯*

School of Pharmacy, Chapman University, Irvine, California, United States of America

ᵒ These authors contributed equally to this work.
* totonchy@chapman.edu

**Data Availability Statement:** Analyzed raw data from this study is available as supplemental material and flow cytometry data will be available at ImmPort under study accession SDY1666. Prior to

## Abstract

Despite 25 years of research, the basic virology of Kaposi Sarcoma Herpesviruses (KSHV) in B lymphocytes remains poorly understood. This study seeks to fill critical gaps in our understanding by characterizing the B lymphocyte lineage-specific tropism of KSHV. Here, we use lymphocytes derived from 40 human tonsil specimens to determine the B lymphocyte lineages targeted by KSHV early during *de novo* infection in our *ex vivo* model system. We characterize the immunological diversity of our tonsil specimens and determine that overall susceptibility of tonsil lymphocytes to KSHV infection varies substantially between donors. We demonstrate that a variety of B lymphocyte subtypes are susceptible to KSHV infection and identify CD138+ plasma cells as a highly targeted cell type for *de novo* KSHV infection. We determine that infection of tonsil B cell lineages is primarily latent with few lineages contributing to lytic replication. We explore the use of CD138 and heparin sulfate proteoglycans as attachment factors for the infection of B lymphocytes and conclude that they do not play a substantial role. Finally, we determine that the host T cell microenvironment influences the course of *de novo* infection in B lymphocytes. These results improve our understanding of KSHV transmission and the biology of early KSHV infection in a naïve human host, and lay a foundation for further characterization of KSHV molecular virology in B lymphocyte lineages.

## Author summary

KSHV infection is associated with cancer in B cells and endothelial cells, particularly in the context of immune suppression. Very little is known about how KSHV is transmitted and how it initially establishes infection in a new host. Saliva is thought to be the primary route of person-to-person transmission for KSHV, making the tonsil a likely first site for KSHV replication in a new human host. Our study examines KSHV infection in B cells extracted from the tonsils of 40 human donors in order to determine what types of B cells are initially targeted for infection and examine how the presence (or absence) of other immune cells influence the initial stages of KSHV infection. We found that a variety of B cell subtypes derived from tonsils can be infected with KSHV. Interestingly, plasma cells

the December 2020 data release, any raw data files can be obtained by request from the corresponding author.

**Funding:** This work was supported by National Cancer Institute grant 5R01239590 to J. Totonchy. The funders had no role in study design, data collection and analysis, decision to publish, or preparation of the manuscript.

**Competing interests:** The authors have declared that no competing interests exist.

(mature antibody-secreting B cells) were a highly targeted cell type. These results lay the foundation for further studies into the specific biology of KSHV in different types of B cells, an effort that may help us ultimately discover how to prevent the establishment of infection in these cells or reveal new ways to halt the progression of B cell cancers associated with KSHV infection.

## Introduction

Kaposi Sarcoma-associated Herpesvirus (KSHV/HHV-8) is a lymphotrophic gamma-herpesvirus. In addition to its role in the pathogenesis of Kaposi Sarcoma (KS) [1], KSHV infection is associated with two lymphoproliferative disorders, multicentric Castleman disease (MCD) and primary effusion lymphoma (PEL) [2,3], as well as a recently characterized inflammatory disorder KSHV inflammatory cytokine syndrome (KICS) [4]. Although KSHV-associated lymphoproliferative disorders are rare, their incidence has not declined as HIV treatment has improved [5,6] suggesting that, in contrast to KS, immune reconstitution is not sufficient to prevent KSHV-associated lymphoproliferative disease in people living with HIV/AIDS. Moreover, the KSHV-associated lymphoproliferative diseases are uniformly fatal with few effective treatment options [7].

Despite the fact that KSHV is lymphotropic and causes pathological lymphoproliferation *in vivo*, study of *de novo* KSHV infection in B lymphocytes has historically been difficult [8]. Resting peripheral B cells and many established B cell-derived cell lines are refractory to KSHV infection but unstimulated tonsil-derived lymphocytes are susceptible to infection [9]. To date, several other groups, including our own, have been successful in infecting B lymphocytes derived from human tonsils [10–15]. KSHV DNA is detectable in human saliva and salivary transmission is thought to be the primary route of person-to-person transmission for KSHV [16–19], making the oral lymphoid tissues a likely site for the initial infection of B lymphocytes in a naïve human host. Thus, in addition to being susceptible to *ex vivo* infection, tonsil lymphocytes represent a highly relevant model for understanding early infection events in KSHV transmission.

The existing studies of KSHV infection in tonsil-derived B cells have explored a limited number of cell surface markers including IgM, immunoglobulin light chains and activation markers on infected cells [10,14,15]. One study using PBMC-derived B lymphocytes identified naïve, memory and plasma cell-like lineages as infection targets in both *in vitro* infection experiments and blood samples from KS patients [20]. However, no studies to date have comprehensively explored the specific B lymphocyte lineages targeted by KSHV infection in human tonsil specimens.

In this study, we performed KSHV infection of 40 human tonsil specimens from diverse donors and utilized lineage-defining immunological markers by flow cytometry to establish the primary B cell lineage tropism of KSHV. Our results demonstrate that the susceptibility of tonsil-derived B lymphocytes to *ex vivo* KHSV infection varies substantially from donor-to-donor, and that a variety of B cell lineages are susceptible to KSHV infection and that, at least at early stages post-infection KSHV is primarily latent in most cell types. In particular, CD138 + plasma cells are highly targeted by KSHV infection despite the fact that they are present at low frequencies in tonsil tissue. We demonstrate that high susceptibility of plasma cells to KSHV infection is not due to the presence of CD138 heparin sulfate proteoglycan as an attachment factor. Moreover, HSPG are not generally important for infection of primary B lymphocytes. Finally, we demonstrate that although the baseline T cell microenvironment does not

seem to influence overall susceptibility of tonsil specimens to KSHV infection, the specific lineage distribution of KSHV infection is affected by the T cell microenvironment and manipulation of CD4/CD8 T cell ratios can alter the targeting of specific B cell lineages by KSHV. These results provide new insights into early events driving the establishment of KSHV infection in the human immune system and demonstrate that alterations in immunological status can affect the dynamics of KSHV infection in B lymphocytes.

## Results

### Variable immunological composition of human tonsil specimens

In order to explore the B cell lineages targeted by KSHV infection in human tonsils, we procured a cohort of 40 de-identified human tonsil specimens from donors of diverse age, sex and self-reported race (Fig 1A and Table 1). Analysis of the baseline frequencies of B and T cell lineages by multi-color flow cytometry (S1 Fig, S1 Table and Table 2) revealed that the composition of individual human tonsil specimens is highly variable (Fig 1B & 1D). This variation was independent of donor age for many lineages. However, overall B cell frequencies declined with age as did germinal center, plasmablast and transitional B cell populations while memory and naïve populations increased in frequency with donor age (Fig 1C). Similarly, most T cell lineages were not significantly correlated with donor age except for CD4+ transitional memory and CD8+ terminal effector lineages which were both moderately, but significantly, negatively correlated with donor age (Fig 1E).

### Variable susceptibility of tonsil-derived B cells to ex vivo KSHV infection

Because of the heterogeneous nature of the tonsil samples, we predicted that each sample would also have variable levels of susceptible B cell subtypes. Therefore, we employed a method for normalizing infectious dose from donor-to-donor in order to obtain cross-sectional data that was directly comparable (Fig 2A). For each sample we used magnetic sorting to isolate untouched naïve B cells, which are a known susceptible cell type [14], and infected 1 million naïve B cells with equivalent doses of cell-free KSHV.219 virus. After infection, bound lymphocytes from the magnetic separation were added back to each sample to reconstitute the total lymphocyte environment. Infected cultures were incubated for three days and analyzed for both B cell lineage markers and the GFP reporter present in the KSHV.219 genome to identify infected cells. We restricted our analysis to a single timepoint at 3 days post-infection in order to observe the establishment of infection in different lineages with minimal contribution of virus-mediated shifting of cellular immunophenotypes, which we observed in our previous study [14]. We first compared the overall lymphocyte populations between baseline (day 0 uninfected), Mock, and KSHV infected conditions to determine whether our infection and culture system and/or KSHV infection itself was causing significant shifts in the B lymphocyte composition of the samples. These results demonstrate that most lineages were not substantially altered by our culture system or KSHV infection, compared to the baseline samples (Fig 2B, S1 Table, S2 Table). However, naïve B cells were significantly reduced compared to baseline levels in both Mock and KSHV infected samples at 3 dpi, suggesting that the infection method may reduce naïve cell survival in this mixed model or that the culture model drives differentiation of naïve cells into a different immunophenotype. Interestingly, B cells with a transitional phenotype are significantly increased compared to baseline in both Mock and KSHV-infected cultures. Given that there is a relationship between transitional (IgD+, CD38mid) and naïve (IgD+, CD38low, CD27-) it is possible that naïve B cells are acquiring increased CD38 expression as a result of the infection and culture process, and are thus falling into the transitional lineage gate at 3 dpi. Finally, this analysis shows that our culture system

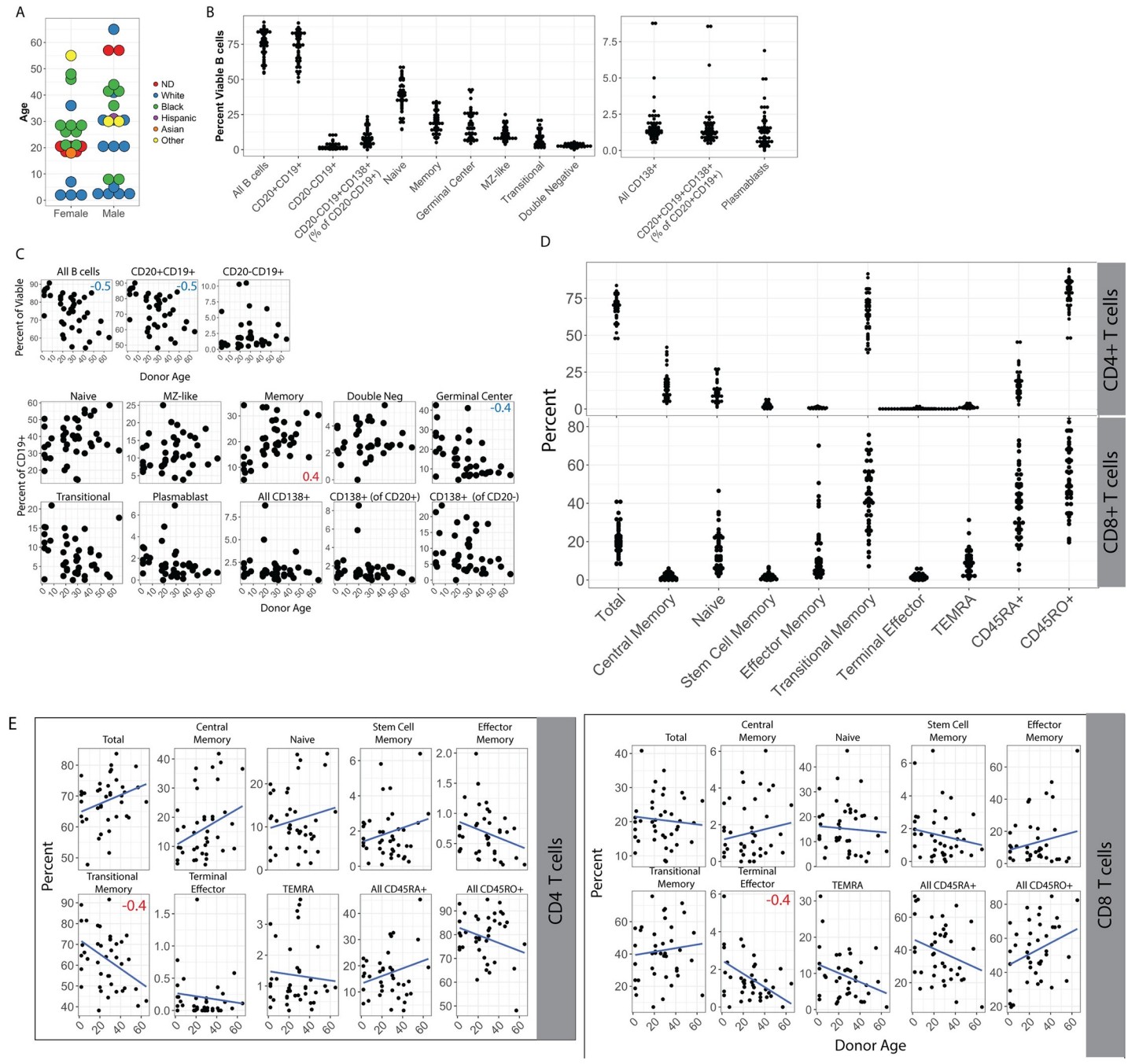

**Fig 1. Variablility and age-dependence of B lymphocyte lineage distribution in human tonsils.** (A) Donor demographics for human tonsil specimens used in this study. Plotted by age (y-axis), sex (x-axis) and self-reported race (color) ND = not determined. (B) Frequency distributions of B cell lineages in tonsil specimens (see S1A Fig and Table 2 for lineage definitions). All B cells are shown as frequency of viable lymphocytes, other B cell subsets are shown as frequency of viable, CD19+ B cells unless otherwise specified in parenthesis. (C) Frequency of B cell lineages based on donor age. Pearson correlation coefficients (r) with an absolute value greater than or equal to 0.4 are shown as red or blue inset text in the subset panels. (D) Frequency distributions of T cell lineages in tonsil specimens (see S1B Fig and Table 2 for lineage definitions). All T cells are shown as frequency of viable CD4+ (top) or CD8+ (bottom) T cell except total CD4+ and total CD8+ (far left) which are shown as frequency of viable lymphocytes. (E) Frequency of CD4+ (left) and CD8+ (right) T cell lineages based on donor age. Pearson correlation coefficients (r) with an absolute value greater than or equal to 0.4 are shown as red or blue inset text in the subset panels.

**Table 1. Donor demographics for the tonsil specimens used in the study (n = 40).**

| Sex, n (%) | |
|---|---|
| Male | 19 (47.5) |
| Female | 21 (52.5) |
| Race, n (%) | |
| ND | 6 (15) |
| White | 17 (42.5) |
| Black | 13 (32.5) |
| Hispanic | 1 (2.5) |
| Asian | 1 (2.5) |
| Other | 2 (5) |
| Age, (years) | |
| (mean ± S.D.) | 26.2 ± 16.47 |
| Range | 2–65 |

does not favor the survival of CD138+ plasma cells indicated by a significant decrease comparing baseline and Mock at 3dpi. However, this effect was lower in the KSHV-infected cultures. This result suggests that KSHV infection is either providing a survival advantage for CD138+ cells in the culture system or driving the differentiation of new CD138+ cells during infection.

Overall, susceptibility of B cells to KSHV infection varied substantially within the cohort with the majority of samples showing between 1 and 2% GFP+ B cells at 3 dpi and an overall range of 0.76–3.27% (Fig 2C, S1 Table). Analysis by point-biserial correlation revealed that

**Table 2. Lineage definitions for lymphocyte subsets used in the study.**

**B Lymphocytes**

| Subset | Molecular Markers |
|---|---|
| Plasma | CD19$^+$, CD20$^{+/-}$, CD138$^{+(Mid\ to\ High)}$, CD38$^-$ |
| Transitional | CD19$^+$, CD138$^-$, CD38$^{Mid}$, IgD$^{+\ (Mid\ to\ High)}$ |
| Plasmablast | CD19$^+$, CD138$^-$, CD38$^{High}$, IgD$^{+\ /-\ (mostly\ -)}$ |
| Germinal Center | CD19$^+$, CD138$^-$, CD38$^{Mid}$, IgD$^-$ |
| Naïve | CD19$^+$, CD138$^-$, CD38$^{Low}$, CD27$^-$, IgD$^{+\ (Mid\ to\ High)}$ |
| Marginal Zone Like (MZ-Like) | CD19$^+$, CD138$^-$, CD38$^{Low}$, CD27$^{+\ (Mid\ to\ High)}$, IgD$^{+\ (Mid\ to\ High)}$ |
| Memory | CD19$^+$, CD138$^-$, CD38$^{Low}$, CD27$^{+\ (Mid\ to\ High)}$, IgD$^-$ |
| Double Negative | CD19$^+$, CD138$^-$, CD38$^{Low}$, CD27$^-$, IgD$^-$ |

**T lymphocytes**

| Subset | Molecular Markers |
|---|---|
| CD4+ | CD19$^-$, CD4$^{+(Mid\ to\ High)}$, CD8$^-$ |
| CD8+ | CD19$^-$, CD4$^-$, CD8$^{+\ (Mid\ to\ High)}$ |
| Naïve | CD19$^-$, CD4+ or CD8+, CCR7$^{+(High)}$, CD45RA$^{+(Mid\ to\ High)}$, CD45RO$^-$, CD28$^+$, CD95$^-$ |
| Stem Cell Memory | CD19$^-$, CD4+ or CD8+, CCR7$^{+(High)}$, CD45RA$^{+(Mid\ to\ High)}$, CD45RO$^-$, CD28$^+$, CD95$^{+\ (Low\ to\ Mid)}$ |
| Central Memory | CD19$^-$, CD4+ or CD8+, CCR7$^+$, CD45RA$^-$ CD45RO$^{+(Mid\ to\ High)}$, CD28$^{+\ (Mid\ to\ High)}$ |
| Transitional Memory | CD19$^-$, CD4+ or CD8+, CCR7$^-$, CD45RA$^-$ CD45RO$^{+(Mid)}$, CD28$^{+\ (Mid\ to\ High)}$ |
| Effector Memory | CD19$^-$, CD4+ or CD8+, CCR7$^-$, CD45RA$^-$ CD45RO$^{+(Mid)}$, CD28$^-$ |
| Terminal Effector Memory | CD19$^-$, CD4+ or CD8+, CCR7$^-$, CD45RA$^-$ CD45RO$^-$, CD28$^-$ |
| TEMRA CD4+ Cells | CD19$^-$, CD4+ or CD8+, CCR7$^-$, CD45RA$^{+(High)}$, CD45RO$^-$, CD28$^-$ |

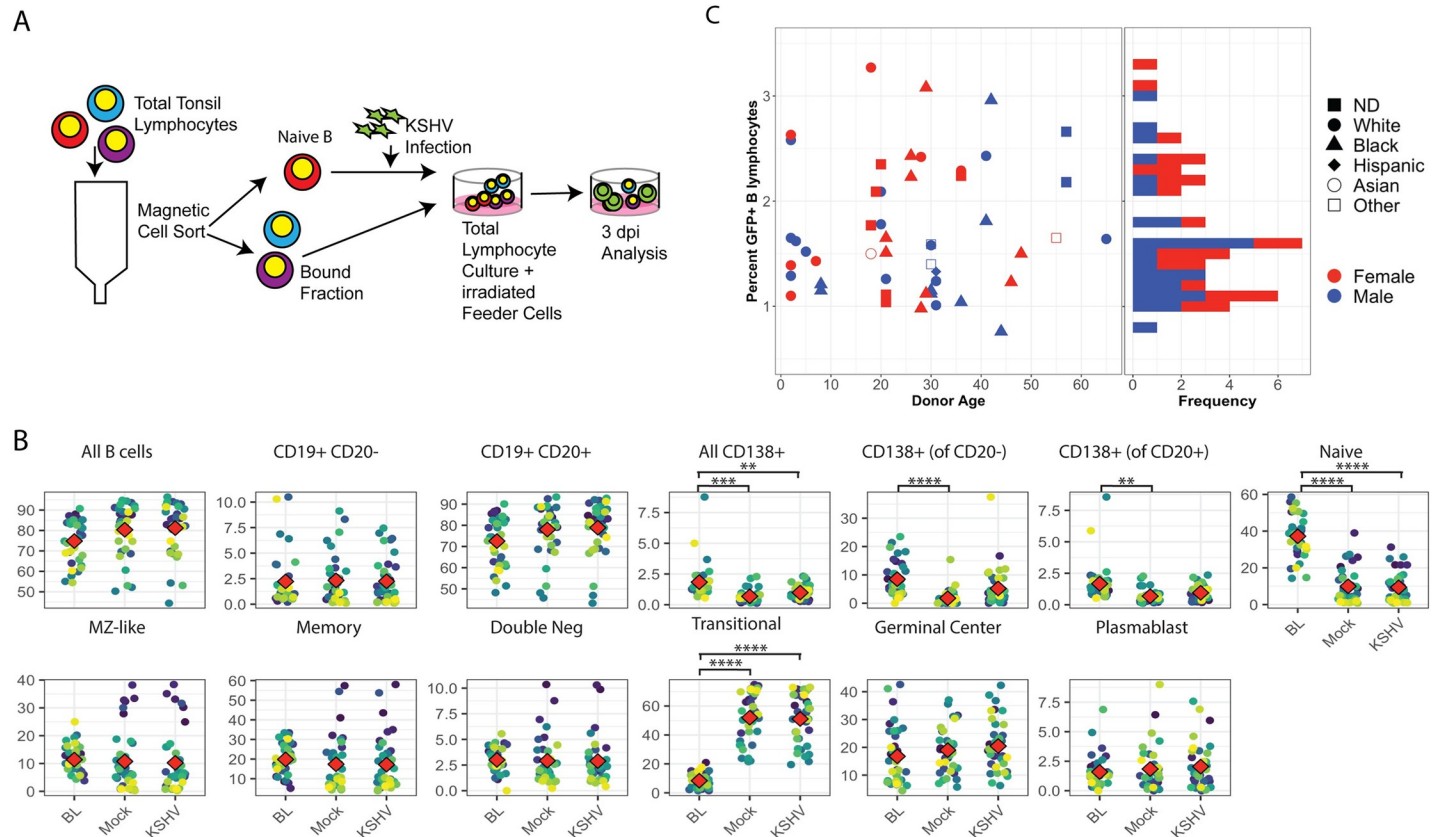

**Fig 2. Tonsil-derived B lymphocytes from diverse donors display variable susceptibility to KSHV infection.** (A) Schematic of lymphocyte infection procedure. Untouched naïve B cells are magnetically separated from total lymphocytes and 1e6 naïve B cells are infected per condition. Following infection, bound fractions are added back to reconstitute a total lymphocyte environment and cells are plated on X-ray irradiated CDw32 feeder cells. Analysis is performed by flow cytometry at 3 days post-infection for B cell lineage markers as shown in S1 Fig and Table 2 and GFP reporter expression for KSHV infection. (B) The effect of KSHV infection and the culture system on overall B cell lineage frequency was examined by comparing the frequency of B cell lineages at baseline (BL, unmanipulated samples) and 3 days post infection in Mock infected and KSHV infected cultures. Student's T test was used to determine statistical significance for all comparisons **p<0.004, ***p<0.0002, ****p<1e-6 (C) Infection frequency data for viable, CD19+GFP+ lymphocytes at 3 dpi n = 50 from 40 tonsil specimens with biological replicates for 10 specimens displayed with respect to donor age (x-axis, left panel), sex (color), and self-reported race (shape, left panel). The histogram in the right panel is included to show the distribution of infection frequencies with the majority of infections resulting in 1–2% GFP+ B lymphocytes at 3 dpi.

susceptibility was not significantly correlated with sex (rpb = 0.17). Kruskal-Wallis rank sum test showed no significant association of race and susceptibility (p = 0.6) and both Pearson (r = 0.09) and Spearman (ρ = 0.01) correlation tests indicated no linear or monotonic relationship between donor age and susceptibility in our data set. Thus, we can conclude that donor demographics do not substantially contribute to the variable susceptibility we observe in our tonsil lymphocyte samples.

## Specific targeting of individual B cell lineages by KSHV infection

We next sought to establish the B lymphocyte tropism of KSHV in human tonsil lymphocytes by determining which B cell lineages are targeted for KSHV infection at early timepoints. Because levels of individual B cell lineages are highly variable between samples (Fig 1B), we have generally represented lineage-specific susceptibility data as the percentage of GFP+ cells within each lineage so that data could be directly compared cross-sectionally within the sample cohort. Our analysis of specific B cell lineages targeted for infection by KSHV revealed that, although they represent a small proportion of the B cells within human tonsils (Fig 1B),

CD138+ plasma cells are infected at high frequencies at this early timepoint. Indeed, CD19+ CD20- plasma cells displayed the highest within-lineage susceptibility of any cell type with several replicates showing 100% infection of this lineage at 3 dpi (Fig 3A). Other B cell lineages were susceptible to infection, but were infected at relatively low within-population frequencies compared to plasma cells (Fig 3A & 3B, S1 Table). Most B cell lineages showed linear correlation between within-lineage infection and overall infection, while others like plasmablast, double negative, and CD20- plasma cells showed no significant correlation between within-subset infection frequency and overall infection (Fig 3B).

The observation that plasma cells are highly targeted for infection is interesting given that we observed a decrease in overall plasma cell numbers in our cultures system that was somewhat abrogated in the KSHV-infected conditions (Fig 2B). Thus, we wanted to determine whether the apparent survival advantage for plasma cells in the KSHV-infected cultures was a direct result of infection. Interestingly, subset analysis for the plasma cells into total, CD20 + and CD20- plasma cells showed that the greatest survival effect (KSHV-Mock for the lineage) was within the CD20- population (Fig 3C, left panel), which was also the population with the highest level of infection among B lymphocytes (Fig 3A) When we plotted survival of each plasma cell sub-population against the percent of infection for that population, there was no significant correlation observed (Fig 3C, right panels). This data supports a conclusion of an indirect effect of KSHV infection on survival or differentiation of a different B cell lineage into CD138+ cells in our KSHV-infected tonsil lymphocyte cultures rather than KSHV conferring a survival advantage only to infected cells.

We next calculated within-subset frequency of infection for each lineage as a proportion of the total B lymphocytes (i.e. within-lineage % GFP x frequency of lineage within B lymphocytes) for each sample in order to determine the contribution of each lineage to the overall infection (Fig 3D & 3E). When shown on a per-sample basis, the data reveals high variability between donors with no discernable contribution of the overall susceptibility (shown by the order of the samples on the x-axis) (Fig 3D). When shown on a per-lineage basis, the data reveals that germinal center, transitional and memory cells make the largest contributions to overall infection, plasma cells and MZ-like cells are intermediate contributors and double negative, naïve and plasmablast lineages make up a minor proportion of the infected cells (Fig 3E).

Next, we wanted to determine whether the targeting of individual B cell lineages by KSHV infection is merely a function of the frequency of that lineage within the sample, or dictated by the virus biology. Pairwise correlations between KSHV infection of specific lineages and the baseline (pre-infection) frequency of that lineage within the sample revealed no significant effect of the baseline frequency of any B cell lineage on the susceptibility of that lineage to KSHV infection, (Fig 3F). Interestingly, there were some significant correlations between baseline frequencies of specific lineages and infection of other lineages. Infection of plasmablasts was positively correlated with the baseline number of both CD20 negative B cells in the culture and the number of CD20+ plasma cells, and infection of CD20+ plasma cells was negatively correlated with the baseline frequency of memory B cells in the sample. Similarly, given that some populations shift in their frequency during the infection timecourse (Fig 2B), we wanted to determine whether KSHV infection of specific lineages was a result of the frequency of that population within the sample at 3 dpi. Pairwise correlations between KSHV infection of specific lineages and the frequency of that lineage at 3 dpi similarly revealed no direct correlations between lineage frequencies and infection frequencies (Fig 3G). These comparisons revealed more strong relationships between lineage frequencies and infection frequencies. Infection of CD138+ plasma cells and CD20+ plasma cells (which are the more numerous of the two plasma cell sub-populations as shown in Fig 1B) was negatively correlated with the total

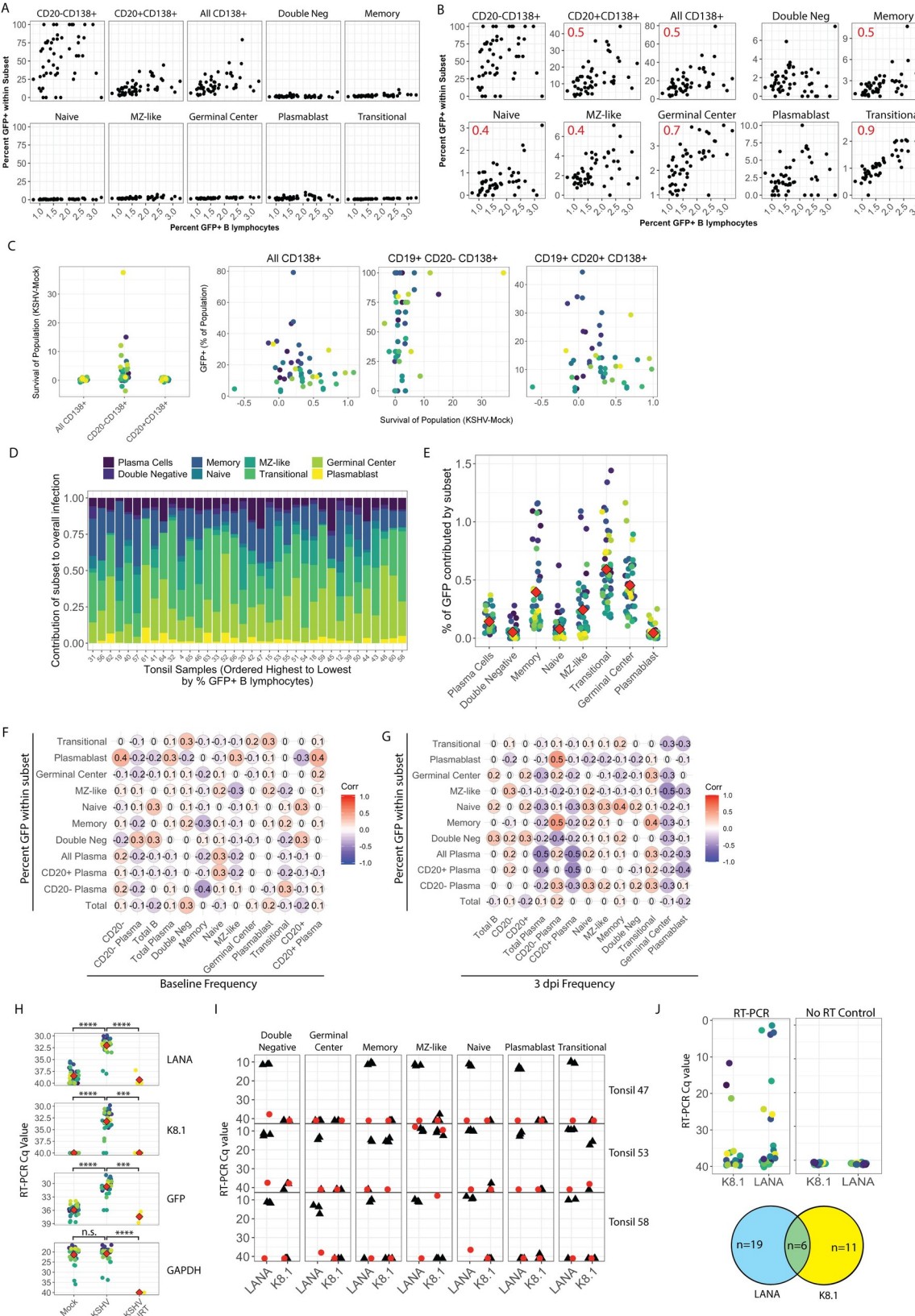

**Fig 3. B lymphocyte lineage tropism of KSHV.** Naïve B lymphocytes from 40 individual human tonsil specimens (n = 50) were infected with KSHV.219 as in Fig 2A. Cells were collected at 3dpi, stained for B cell lineages as shown in S1A Fig and Table 2 and analyzed flow cytometry for B cell lineages and KSHV infection based on GFP reporter expression. The within-lineage infection frequency (y-axis) as a function of overall B cell infection frequency (x-axis) at 3dpi in for n = 50 infections from 40 individual tonsil specimens shown in (A) normalized to 100% or (B) scaled to each individual lineage population. Pearson correlation coefficients (r) with an absolute value greater than or equal to 0.4 are shown in (B) as red inset text in the subset panels. (C) The survival of plasma cell lineages (frequency of viable B lymphocytes in KSHV-Mock samples at 3 dpi) was plotted for all CD138+, CD20-CD138+ and CD20+CD138+ lineages (left panel) and each lineage's frequency of KSHV infection was plotted against its survival (right three panels) with individual tonsil samples designated by color. Statistical analysis was performed for both linear (Pearson) and monotonic (Spearman) correlations. For all plasma cells r = 0.15, ρ = 0.2; for CD20- plasma cells r = 0.3, ρ = 0.32, for CD20+ plasma cells r = 0.15, ρ = 0.13. The contribution of specific B cell lineages to overall infection in each sample was calculated as % GFP within lineage * % of lineage within viable B cells in each sample, and results are shown in (D) for each tonsil sample ordered from highest to lowest (right to left) based on overall GFP+ B lymphocytes (overall susceptibility) and (E) by lineage with individual samples designated by color and the mean infection frequency of each lineage shown as a red diamond. Correlation matrix analysis showing linear relationships (Pearson's correlation coefficient) between within-lineage infection frequency and (F) baseline (pre-infection) overall frequency or (G) 3 dpi overall frequency in KSHV-infected cultures for B lymphocytes and their subsets. Statistical power analysis indicates that this dataset can predict correlations at the level of r > |0.4| with alpha = 0.05 and power = 0.8. Thus, r values ≥ 0.4 can be considered a statistically significant correlation for this data. (H) KSHV transcripts LANA and K8.1 as well as GFP and GAPDH transcripts analyzed for n = 20 tonsil specimens by RT-PCR in bulk lymphocyte cultures at 3 dpi. Left panels show Ct values with individual samples designated by color and the mean Ct values for each condition shown as a red diamond. Student's T-test was used to determine statistical significance for all comparisons ***p<1e-7, ****p<2e-10. (I) At 3 dpi, 10 million KSHV-infected lymphocytes from three tonsil specimens were stained for B lymphocyte surface markers and lineages were sorted into Trizol LS. RNA was extracted, reverse transcribed (black triangles) or amplified without reverse transcriptase (NRT, red circles) and analyzed by nested RT-PCR for viral transcripts in sorted B lymphocyte lineages. (J) At 3 days post-infection, 1 million KSHV-infected B cells from three tonsil specimens were stained for viability, CD19 and CD138. 187 single cells that were viable, CD19+, CD138+ were sorted into 96-well plates and analyzed by nested RT-PCR for viral transcripts. Colors indicate single plasma cells analyzed with RT-PCR (left) or control reactions including no reverse transcriptase (right) The bottom panel is a venn diagram quantitating the number of plasma cells in RT-PCR reactions which amplified for each and both viral transcripts.

population of plasma cells and the CD20+ sub-population of plasma cells. This result may indicate that infection of CD20+ plasma cells results in significant toxicity for that lineage. Moreover, infection of many B cell lineages was negatively correlated with germinal center B cells at 3 dpi with MZ-like B cell infection being significantly associated. This observation could indicate that lymphocyte cultures with a microenvironment that favors the survival of germinal center cells establish a different subset distribution of B lymphocyte infection. Finally, the 3dpi level of CD20- plasma cells was positively correlated with infection of both plasmablast and memory B cell lineages. This data could indicate that these lineages are differentiating into plasma cells upon infection. Taken together these data indicate that the B lymphocyte tropism of KSHV is broad and highly variable from donor to donor. Plasma cells are highly targeted as a lineage, but transitional, memory and germinal center lineages make up the bulk of the viral load in tonsil specimens. The distribution of KSHV infection among lineages is not simply a function of lineage population frequency within the sample, and is likely dictated by cell-intrinsic factors as well as complex immunological interplay within the sample that remains to be fully characterized.

## Viral gene expression in KSHV infected B lymphocytes

Our observation that KSHV targets diverse B cell lineages for infection is based on expression of the GFP reporter in the KSHV.219 genome, which is controlled by a non-viral EF1-alpha promoter. We wanted to validate that the GFP signal we observe by flow cytometry represents *bona fide* infection and not simply virus entry. In order to do this, we first examined total RNA extracted from mock or KSHV-infected lymphocyte cultures at 3 dpi and analyzed these samples by RT-PCR for viral transcripts (LANA and K8.1) as well as GFP as a marker for virus entry and GAPDH as a housekeeping gene for the efficacy of RNA extraction (Fig 3H). The data show that viral transcripts and GFP are absent in Mock samples but present in KSHV-infected samples with an average -ΔCt of 6.1 cycles for LANA, 6.6 cycles for K8.1 and 4 cycles

for GFP. NRT controls were consistently negative, confirming that viral DNA was not the source of genetic material for these results. Both LANA and K8.1 were detected in the majority of samples, suggesting a mix of lytic and latent infection programs in B lymphocytes. This data demonstrates *bona fide* infection, with the production of viral transcripts, is present in tonsil lymphocyte samples at 3 dpi.

Given that the bulk RT-PCR data showed mixed lytic and latent transcripts present in our tonsil lymphocyte cultures, we wanted to determine whether B lymphocyte lineages preferentially undergo a particular viral replication program. To accomplish this, we performed large-scale lymphocyte infections, as above, for three unique tonsil specimens, stained for B cell lineage markers, and sorted individual lineages using our FACS Aria Fusion cell sorter. We were able to obtain between 10,000 and 200,000 cells for each lineage from the cell sorting. Total RNA was extracted from sorted samples and subjected to nested RT-PCR analysis for GAPDH, LANA and K8.1. These results show that LANA transcripts are present in all lineages for at least 2/3 tonsil samples analyzed. Transcription of the K8.1 late lytic gene was observed in memory cells and transitional cells for 1/3 tonsil samples along with LANA transcripts indicating that in this sample there was a mixture of lytic and latent cells within the lineage populations. NRT negative controls (red circles) for most lineages were negative or amplified >10 cycles later than the matched RT positive samples indicating that viral DNA was not the source of genetic material for these results. However, the MZ-like lineage had positive amplifications in NRT controls for 2/3 tonsil samples. This result may indicate that in some samples there is a high load of KSHV DNA present in this lineage so that even the extensive DNase digestion used in our protocol failed to remove it sufficiently (Fig 3I). Because plasma cells are a low abundance cell type in our tonsil samples, we were uncertain whether bulk sorting would result in sufficient cells to successfully extract RNA for RT-PCR analysis. Thus, in order to determine the viral transcription program in plasma cells we gated viable/CD19+/CD138+ cells and sorted single cells directly into 96-well plates containing a hypotonic lysis buffer. We performed single cell nested RT-PCR analysis without RNA extraction for LANA, K8.1 and GAPDH on 187 plasma cells from three unique tonsil samples (Fig 3J, left panel). NRT controls were consistently negative confirming that viral DNA was not the source of genetic material for these results (Fig 3J, right panel). We observed 19 plasma cells expressing LANA transcripts only, 11 plasma cells expressing K8.1 only and 6 plasma cells expressing both viral transcripts (Fig 3J, bottom).

Our gene expression data validates the lineage-specific tropism observed in our flow cytometry data, showing that each lineage identified as susceptible by GFP expression also contains viral transcripts, indicating *bona fide* KSHV infection. Moreover, these results demonstrate that most B cell lineages express latent transcripts only, with few lineages including plasma cell, memory, transitional and possibly MZ-like lineages contributing to lytic replication. Given that transitional and memory cells represent a high proportion of the per-sample viral load in our analysis (Fig 3D & 3E) it is not surprising to find that they are competent for lytic replication. However, germinal center cells are uniformly latent in this data, but were a highly represented cell type in our analysis of lineage contributions to overall infection. This may indicate that germinal center B cells are another highly targeted cell type. Finally, differences in this data between tonsil samples indicates that lytic replication in our system may be more dependent upon host factors than lineage-specific factors.

## KSHV infection of B lymphocytes does not rely on heparin sulfate proteoglycans

Previous studies have shown that heparin sulfate proteoglycans (HSPG) of the syndecan family can serve as an attachment factor facilitating KSHV entry via interaction with the gH/gL

glycoprotein complex [21]. In order to test whether the high susceptibility of tonsil-derived plasma cells was due to increased attachment via CD138 (syndecan-1), we attempted to selectively neutralize KSHV entry by pre-treating cell-free virus particles with soluble recombinant CD138 (srCD138) protein prior to infection. We utilized recombinant CD138 for these experiments rather than a neutralizing antibody against CD138 because the biochemistry of the putative interaction between CD138 and gH has not been established. Thus, the soluble protein will contain all of the protein sequences that might be bound by gH while an antibody blocks only specific epitopes which may or may not be part of the interaction domain. Pre-treating KSHV virions with 12.5μg/ml of srCD138 showed a small decrease in overall KSHV infection of B lymphocytes in 6 of 7 tonsil samples tested (Fig 4A). However, the inhibition was not dose-dependent for any sample. B cell lineage analysis revealed decreased infection of plasma cell lineages in 3 of 7 samples, but again the effect was inconsistent within the data set and was not dose-dependent for any sample. KSHV infection of other B cell lineages was similarly inconsistently affected by srCD138 treatment of virus particles (Fig 4B). Next, we used srCD138 pre-treated virions to infect human fibroblasts to determine whether srCD138 was able to neutralize gH on another cell type. In these experiments we observed a slight decrease in infection in 4 of 6 replicates (Fig 4C). Taken together, these results suggest that, although srCD138 treatment seems to weakly neutralize KSHV viral particles, the effect is not B cell specific. As a way of confirming these results with a cell-directed method as opposed to a virus neutralization approach, we used heparinase treatment to remove all HSPG prior to KSHV infection. Treatment of human fibroblasts with a heparinase I/III blend resulted in decreased cell surface heparin sulfate by flow cytometry analysis (Fig 4D) and, as demonstrated previously in human fibroblasts [22], consistently reduced KSHV infection of treated target cells compared to untreated controls (Fig 4E). Lymphocytes had lower steady-state HSPG levels compared to fibroblasts, which was further reduced by heparinase treatment (Fig 4F). Interestingly, heparinase treatment of lymphocytes did not result in a reproducible decrease in KSHV infection (Fig 4G). We next wanted to determine whether there was any effect of heparinase treatment on KSHV infection of particular B cell lineages. Because the most reliable cell surface marker for plasma cells is the CD138 HSPG, which would be removed by heparinase treatment, we first confirmed that plasma cells recovered CD138 expression before our 3 dpi analysis timepoint (Fig 4H). Subset analysis revealed no significant differences in KSHV infection between control and heparinase treated populations for any B lymphocyte lineage (Fig 4I). Taken together these data do not support the conclusion that plasma cell-expressed CD138 is used as an attachment factor for KSHV entry and, indeed, indicates that HSPG are not a significant factor in KSHV attachment to B primary lymphocytes.

## Immune status alters KSHV infection of B lymphocytes

KSHV lymphoproliferative disorders occur primarily in the context of immunosuppression, and other studies have shown interactions between T cells and KSHV infected B cells in tonsil lymphocyte cultures affecting the frequency of lytic reactivation [11]. Therefore, we wanted to determine whether the immunological composition of the tonsil lymphocyte environment would affect the establishment of KSHV infection in B lymphocytes and specifically whether overall susceptibility or targeting of particular B cell lineages is influenced by the presence or absence of T cells. Like B cell lineages, levels of CD4+ and CD8+ T cell lineages vary considerably between tonsil donors (Fig 1D). However, unlike B lymphocyte subsets, the distribution of T cell subsets are not generally correlated with donor age (Fig 1E). Examination of whether the ratio of CD4/CD8 T cells in individual tonsil specimens was correlated with the susceptibility of B lymphocytes to KSHV infection revealed no significant correlation (Fig 5A). Next, we

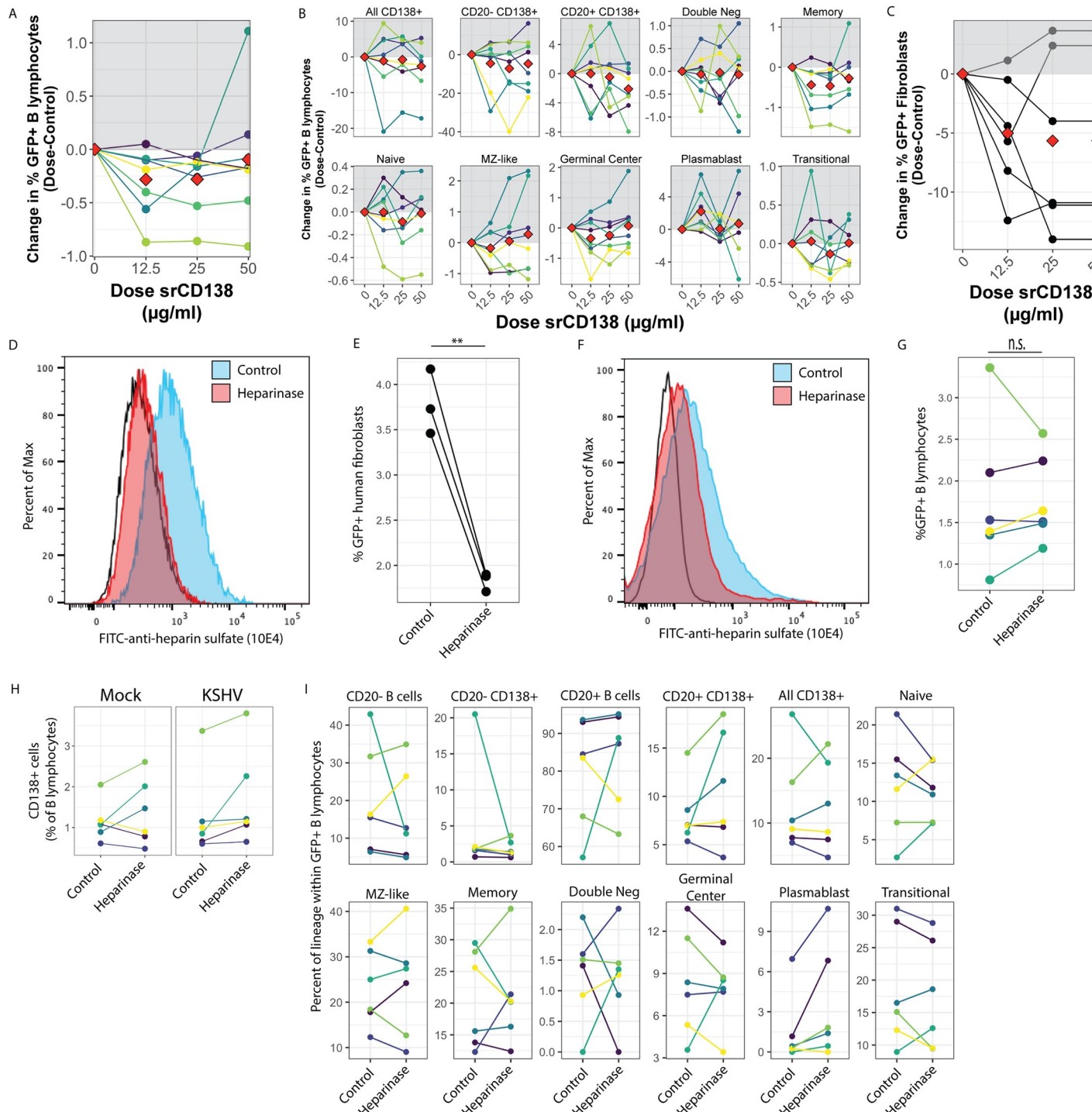

**Fig 4. CD138 and heparin sulfate proteoglycans as attachment factors for KSHV in B lymphocytes.** (A) Purified KSHV virions were pre-treated with srCD138 protein at indicated concentrations (x-axis) and used for infection of B lymphocytes. Cells were collected at 3dpi, stained for B cell lineages as shown in S1A Fig and Table 2 and analyzed by flow cytometry for lineage frequencies and KSHV infection by GFP reporter expression. 8 experimental replicates of 5 unique tonsil specimens are shown where the average infection rate was 1.8±0.5% in untreated controls. Data is represented as change in GFP+ cells at each dose of srCD138 compared to untreated control. Colors denote individual experimental replicates and red diamonds are the mean change at each dose for all replicates. (B) Data as in (A) for within-subset GFP quantitation. (C) Similar experiments were performed in E6/E7 transformed fibroblasts derived from human tonsils. 6 experimental replicates are shown where the average infection rate was 25.5±16% in untreated controls. Data is represented as change in GFP+ cells as in (A). (D) 1 million E6/E7 transformed human tonsil fibroblasts were treated with 4.5 units of heparinase I/III blend for 24 hours and removal of heparin sulfate proteoglycans was verified by flow cytometry using a heparin sulfate-FITC antibody. Black line indicates no antibody control. (E) Control (untreated) or heparinase-treated fibroblasts were infected with KSHV.219 and

analyzed for infection by GFP reporter expression at 3 dpi. Student's T-test was used to compare control and heparinase-treated cultures for n = 3 experimental replicates p = 0.007. (F) 25 million human tonsil lymphocytes were treated with 9U of heparinase I/III blend and plated on X-ray irradiated CDW32 feeder cells for 24 hours. After incubation removal of heparin sulfate proteoglycans was verified by flow cytometry as in (D). (G) After heparinase treatment lymphocytes were fractionated and infected as shown in Fig 2A and viable, GFP+ B lymphocytes were quantitated by flow cytometry at 3 days post-infection. 6 experimental replicates with 6 tonsil specimens were performed and colors designate unique samples and can be compared between this panel, panel H and panel I. Student's t-test was performed to compare infection of control and heparinase treated lymphocytes p = 0.969. (H) the recovery of cell surface CD138 HSPG after 3 days of culture was examined by comparing CD138+ cells as a percent of viable B cells in control and heparinase treated samples for both mock and KSHV-infected conditions. Colors indicate unique tonsil specimens. Student's T-test was performed to compare control vs. heparinase conditions for mock p = 0.57, for KSHV p = 0.52 (I) Data for KSHV-infected cultures at 3dpi with or without heparinase pre-treatment as in (G) showing the level of KSHV infection for specific B cell lineages. Student's T-test indicates no significant difference comparing control and heparinase treated samples for any lineage.

examined correlations between B cell infections and baseline levels of various CD4 and CD8+ T cell subsets to determine whether any T cell lineages affected the tropism of KSHV for particular B cell lineages (Fig 5B). Interestingly, levels of naïve and stem cell memory CD4+ T cells was

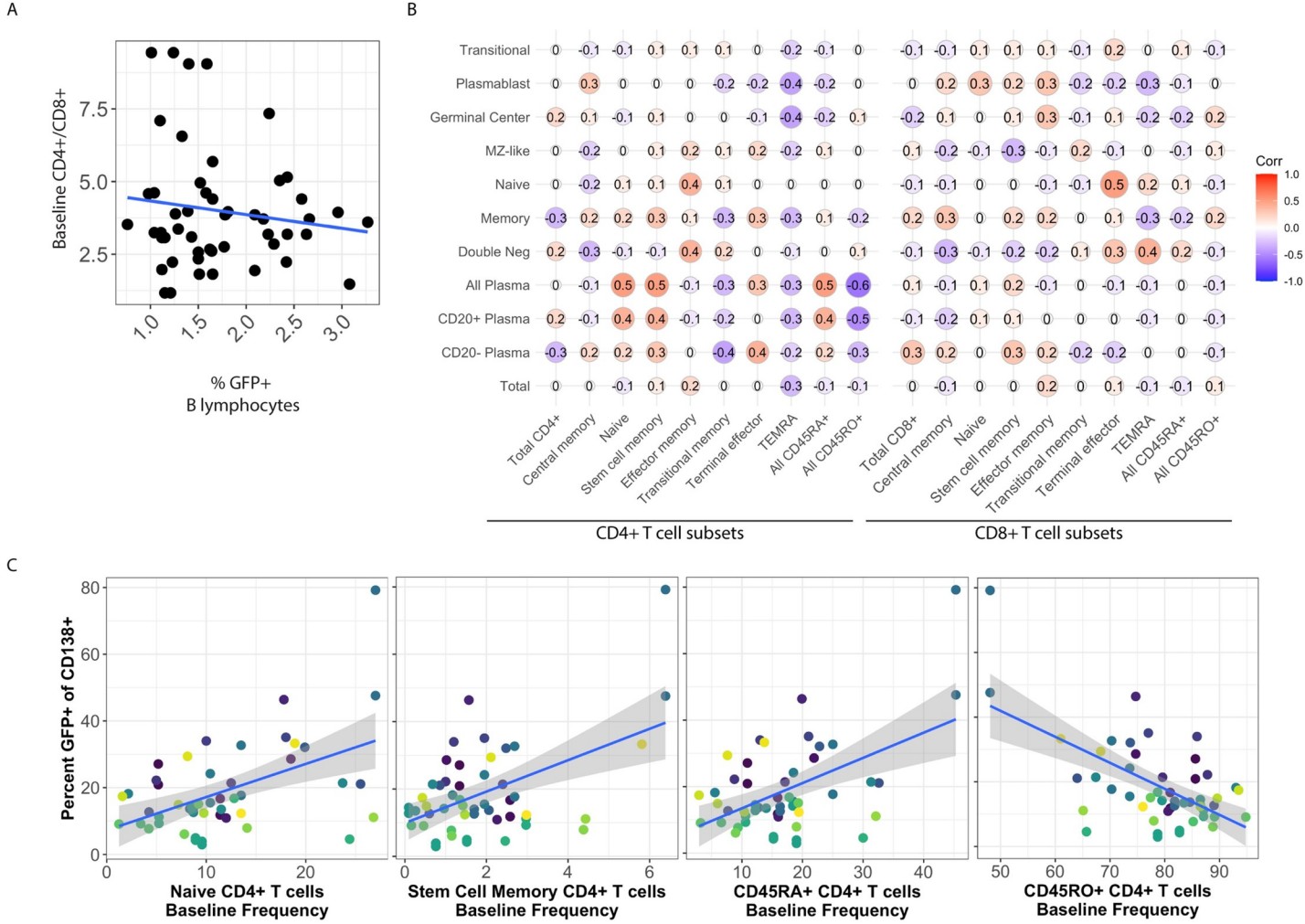

**Fig 5. The donor-specific CD4+ T cell microenvironment influences infection of CD138+ plasma cells.** (A) For each donor, baseline levels of CD4+ T cells/CD8+ T cells is plotted against the overall susceptibility of the specimen based on the percent of GFP+ B lymphocytes at 3 dpi. Blue line indicates least means linear regression. Correlation analysis reveals no significant linear or monotonic relationship between the variables. (B) Pairwise linear correlations (Pearson method) were performed for overall (Total) and lineage-specific KSHV infection at 3 dpi (vertical axis) and baseline T cell subsets as defined in S1 Fig and Table 2. Power analysis reveals that the data set can predict significant correlations at the level of $p > |0.4|$ with alpha = 0.05 and power = 0.8. (C) Scatter plots of significant correlations from (B) between the baseline frequency of CD4+ T cell subsets (x-axis) and KSHV infection of CD138+ plasma cells (y-axis). Blue lines indicate least means linear regression and grey shading is standard error. Colors indicate unique tonsil specimens and can be compared across panels within the figure.

positively correlated with infection of plasma cells (Fig 5B and Fig 5C, left panels) with a greater effect on CD20+ plasma cells than CD20- plasma cells, while overall levels of CD45RO+ activated memory T cells were negatively correlated with KSHV infection of plasma cells (Fig 5C, right panel). In addition, although the correlations were too weak to be significant in this data set, nearly every B cell lineage and overall KSHV infection was negatively correlated with the presence of CD4+ T cells expressing a TEMRA phenotype (Fig 5B). Baseline levels of CD8+ T cells had less effect on KSHV infection with only one significant positive correlation observed in our dataset between CD8+ terminal effector cells and KSHV infection of naïve B cells (Fig 5B).

To determine whether manipulating the T cell environment would affect KSHV infection in individual tonsil samples, we performed T cell depletion experiments. Based on the correlation data shown in Fig 5B, we hypothesized that depletion of CD4+ T cells would have a greater effect on KSHV infection. We performed KSHV infections in which total lymphocytes, CD4-depleted total lymphocytes or CD8-depleted total lymphocytes were added back following infection of sorted naïve B cells. At 3dpi, we validated T cell depletions (Fig 6A) and analyzed KSHV infection of B lymphocytes. The effect of T cell depletion varied substantially from sample to sample and neither CD4 nor CD8 depletion significantly altered overall KSHV infection in tonsil-derived B cells when data from 11 tonsil samples were aggregated (Fig 6B). However, due to the heterogeneous nature of our tonsil samples (Fig 1B & 1D), we hypothesized that the baseline T cell composition of each sample might influence the effect of T cell depletion on KSHV infection. Indeed, when the change in KSHV infection in depleted fractions is plotted against the baseline CD4/CD8 T cell ratio, we observe that KSHV infection increased when depletions were performed in samples with high baseline levels of CD4+ T cells (Fig 6C). We next analyzed the effect of T cell depletion on KSHV infection of specific B cell lineages (Fig 6D). These data reveal that depletion of CD4+ T cells increases infection of plasma cells as well as MZ-like and Transitional B cell lineages and that the effect is dependent upon the baseline CD4/CD8 T cell ratio in the sample with CD4 + T cell-rich samples showing the greatest effect. Interestingly, CD8 depletions altered infection of different lineages compared to CD4 depletions, but showed a similar dependence on the baseline level of CD4+ T cells. Power analysis indicates that only the effect on CD138+ cells and MZ-like lineages in the CD4-depleted condition and plasmablast lineage in the CD8-depleted condition can be considered statistically significant based on the sample size. Taken together, these data support the conclusion that the T cell microenvironment influences the lineages targeted by KSHV infection in the B lymphocyte compartment.

## Discussion

*Ex vivo* infection of tonsil lymphocytes is emerging as a viable strategy for studying early infection events for KSHV infection in B lymphocytes. However, the existing literature on KSHV infection of tonsil lymphocytes is highly varied in both approach and outcome. Hassman et. al. used cell free wild-type BCBL-1 derived KSHV virions to infect total CD19+ B lymphocytes from tonsil specimens and used staining for LANA as the only marker for infection, thus limiting their analysis to latently infected cells [10]. Bekerman et. al. also used isolated CD19+ B cells as infection targets, but employed a co-culture infection procedure using iSLK cells infected with the recombinant KSHV.219 strain employing the GFP reporter as a marker for infection. Nicol et. al. also employed a co-culture method to infect tonsil lymphocytes with KSHV.219 but used Vero cells as producers and did not isolate CD19+ B cells prior to infection. In two studies, Myoung et. al. used cell free KSHV.219 produced from Vero cells to infect mixed lymphocyte cultures. With the exception of Bekerman et. al, all of the above-mentioned studies employed some kind of activating agent (PHA stimulation or CD40L stimulation) to manipulate the activation and/or proliferation of cells *in vitro*. For our studies, we used cell-

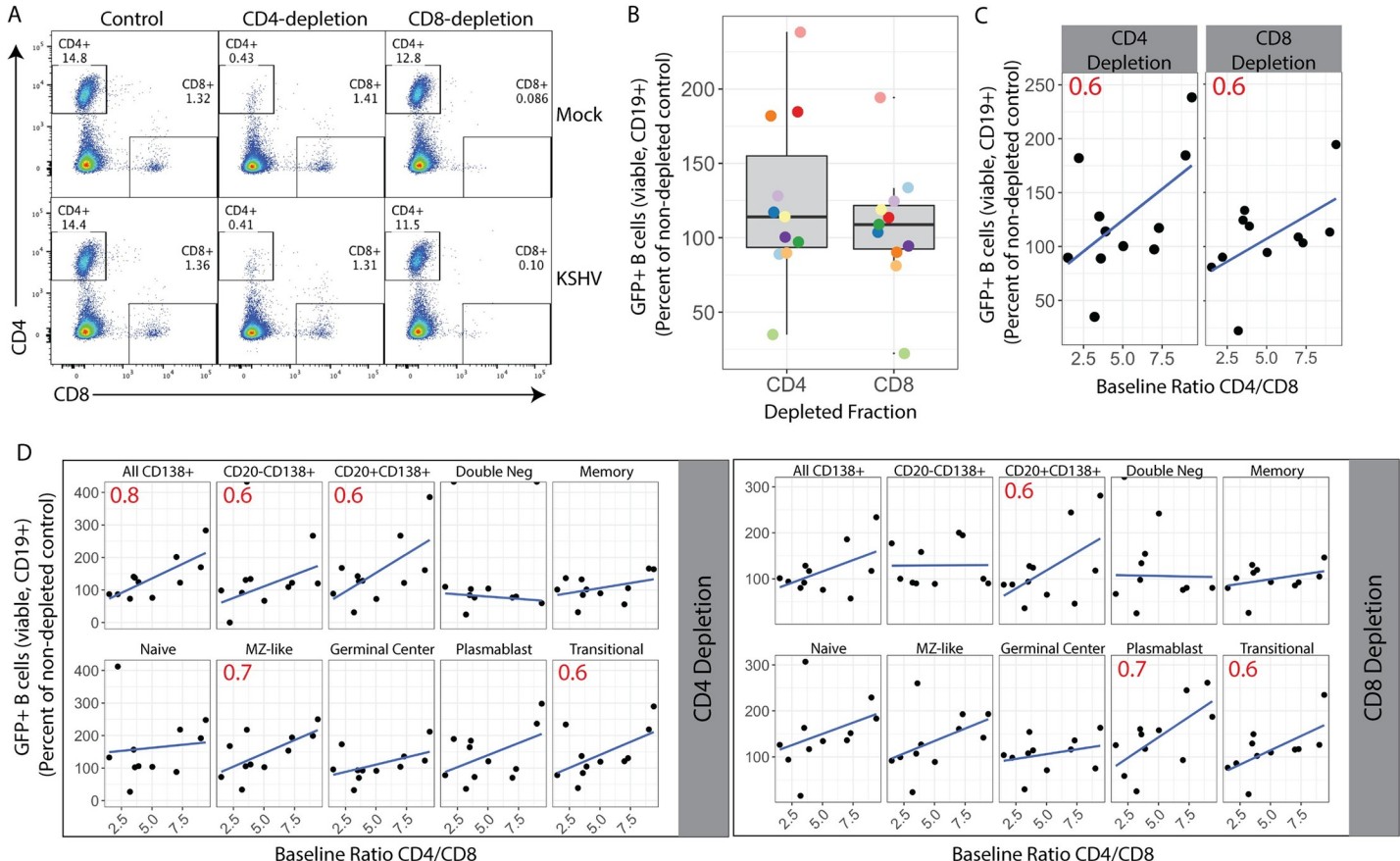

**Fig 6. Manipulation of T cell microenvironment alters KSHV tropism for B cell lineages.** Naïve B lymphocytes from 11 human tonsil specimens were infected with KSHV.219 as in Fig 2A, total lymphocytes added back following infection were either untreated or depleted of CD4+ or CD8+ T cells. At 3 days post-infection cells were collected. (A) T cell depletions were validated by flow cytometry and (B) Cells were stained for B cell lineages as shown in S1A Fig and Table 2 and analyzed flow cytometry for lineage frequencies and KSHV infection by GFP reporter expression to determine the change in GFP+ B lymphocytes (viable, CD19+) in CD4 or CD8 depleted samples compared to non-depleted controls. Colors indicate individual tonsil specimens (n = 11). Student's T test reveals no statistically significant change in GFP+ B cell frequency between either CD4-depleted or CD8-depleted and non-depleted controls. (C) infection data for T cell depletion studies as in (A) plotted against the baseline (pre-infection) CD4/CD8 T cell ratio in the sample. (D) within-lineage infection frequency plotted against baseline (pre-infection) CD4/CD8 T cell ratio. For B and C linear regression of the data is shown as a blue line and Pearson correlation coefficients (r) are shown as red text within panels only for r with an absolute value ≥ 0.6. Statistical power analysis indicates that this dataset can predict correlations at the level of r > |0.7| with alpha = 0.05 and power = 0.8. Thus, r values ≥ 0.7 can be considered a statistically significant correlation for this data.

free, iSLK-derived KSHV.219 to infect naïve B lymphocytes followed by reconstitution of the total lymphocyte environment. We also avoided activation of lymphocytes in both the isolation and culture procedures using our previously-characterized CDw32 feeder cell system [14]. To date, no consensus has yet emerged on how to perform tonsil lymphocyte infection studies with KSHV, and how differences in infection and culture procedure influences the resulting data remains to be established.

Although previous studies in mixed lymphocyte cultures have explored limited surface markers including immunoglobulins and activation markers [10,15] and NK cell ligands [13], these studies essentially treated all B cells as one population. Targeting of specific lineages including naïve, memory and CD138+ plasma cell-like B lymphocytes was explored by Knowlton et. al. using B cells derived from peripheral blood, but their detection method for infected cells was immunostaining for ORF59 protein and thus their enumeration of infection was biased towards cells undergoing lytic replication [20]. Our current study is the first to use a

comprehensive panel of lineage-defining cell surface markers to carefully explore the lineage-specificity of KSHV infection in B lymphocytes.

Although previous studies of KSHV infection in tonsil-derived B lymphocytes have shown the acquisition of plasmablast-like features at later timepoints post-infection [8,10], the CD38 high plasmablast lineage is a minor proportion of our infected cultures at 3 dpi. Based upon these studies, we might expect to see the emergence of more plasmablast-like cells over time and it will be interesting to determine whether this is a result of infected plasmablasts expanding or trans-differentiation from other lineages. Our observation that KSHV-infected B cells are primarily latent in our mixed tonsil lymphocyte cultures is consistent with previous observations that T lymphocytes control lytic reactivation of KSHV in tonsil-derived B cells [11].

For the gamma-herpesviruses EBV and MHV68, the current consensus is that naïve B cells are the primary infection target and infected cells transit the germinal center as a way of increasing viral load without resorting to lytic replication and lifelong latent infection is established in memory B cells while plasma cells are a source of lytic replication constantly replenishing the viral reservoir by producing virus which infects more naïve B cells [23]. Here, we show that multiple lineages from human tonsils, including terminally differentiated CD138 + plasma cells, can be targeted by KSHV for *de novo* infection. In our data, plasma cells are not primarily lytic as in EBV models, but instead are a mix of lytic and latent transcription programs with latency slightly predominating (Fig 3J). This result is not surprising given that KSHV-associated lymphoproliferative diseases are characterized by primarily latent infection in cells with plasma-like features but PEL derived cell lines are competent for KSHV replication given the proper stimulus. In future studies, it will be interesting to examine what factors influence the lytic/latent balance for KSHV in primary plasma cells. Additionally, our data indicate that memory cells are competent for lytic replication (Fig 3I), which is also different from the current model for EBV. However, the studies upon which the EBV models are based are primarily examination of cells from previously infected human hosts, representing EBV distribution in an established infection. We are unaware of any study of EBV infection in human tonsil that recapitulates the early infection timepoint and the level of detail in characterizing B cell lineages that is used here, thus it is difficult to place our data in the context of EBV infection. Given that recombinant EBV molecular genetics systems and *ex vivo* tonsil lymphocyte culture systems are now well established, perhaps a second look at the early stage B lymphocyte tropism of EBV is warranted.

Our finding that KSHV efficiently targets CD138+ plasma cells early in infection of tonsil B lymphocytes is particularly intriguing and relevant in the context of KSHV-mediated lymphoproliferative diseases, which often have a plasma cell or plasmablast-like phenotype [24,25]. Particularly for PEL, which uniformly presents as a clonal CD138+ neoplasm, these results suggest that the pathological cells may not be derived from KSHV-driven differentiation from less mature lineages, but instead could be the result of modifications of differentiated plasma cells by direct infection. Recent studies have revealed that XBP-1, a critical cellular mediator of the unfolded protein response (UPR) which is essential for the differentiation of plasma cells [26], can activate expression of KSHV vIL-6 without inducing ORF45 or other lytic genes [27]. The fact that the UPR and XBP-1 are uniformly active in immunoglobulin-producing cells, like plasma cells, suggests that these cells may provide a unique niche for KSHV persistence where vIL-6 can be produced to support infected plasma cell survival in the absence of KSHV lytic replication. Future studies will examine whether vIL-6 is responsible for the survival advantage we observed for plasma cells in the KSHV-infected conditions in this study (Fig 2B).

Current models of human plasma cell maturation suggest that CD20 expression is lost on plasma cells as they mature and migrate from peripheral lymphoid organs (such as the tonsil) to the blood and finally the bone marrow [28]. Thus, CD138+CD20- plasma cells in tonsil that

are highly targeted by KSHV in our analysis may represent a population of cells that is ready to leave the tonsil and migrate to the bone marrow. A few studies have shown KSHV infection in bone marrow from patients with MCD [29,30] and HIV positive patients without MCD [31], and it would be interesting to pursue the idea that KSHV uses plasma cells to disseminate to the bone marrow early in infection. Certainly, our results highlight the virology of KSHV in primary plasma cells as an area urgently requiring further study.

In this study, we were unable to establish that targeting of plasma cells was due to enhanced virion binding via gH/gL interaction with the CD138 HSPG molecule, as was suggested by a previous study [21]. However, we acknowledge that selectively blocking virion binding to a specific HSPG in a primary cell system is technically difficult, and we have no way to directly verify that our neutralization of gH/gL binding sites using soluble CD138 was effective. Our data using heparinase treatment to remove HSPG from B lymphocytes prior to infection supports the conclusion that CD138 is not used as an attachment factor for plasma cells, and indeed, HSPG are generally dispensable for infection of B cells in our system (Fig 4G–4I). These results are consistent with a previous study that shows low HSPG levels on the KSHV-susceptible MC116 lymphoma cell line [32] and another study showing that ectopic HS expression enhanced binding but was not sufficient to allow efficient KSHV infection of the BJAB lymphoma cell line [33]. Thus, the mechanisms underlying KSHV targeting of plasma cells and other B lymphocyte lineages for infection remains to be established. Additional studies in lymphoma cell lines have identified Ephrin receptors as critical factors in KSHV entry [34,35]. Thus, our future studies in this area will examine Ephrin family receptors in KSHV infection of tonsillar B lymphocyte lineages.

Our characterization of immunological diversity of a large cohort of human tonsil specimens will be of interest to the general immunology community. Although a few studies have examined T cell [36] or B cell [37] subsets in tonsils associated with particular disease states. To our knowledge, only one other study has used multicolor flow cytometry to examine the immunological composition of both B cells and T cells in a large cohort of human tonsil samples [38], and this study was focused on comparing the microenvironments present in matched tonsils and adenoids rather than comparison between donors based upon demographic data as we have done here.

In this study, we make the observation that manipulating the T cell composition has a more profound effect on KSHV infection in tonsil specimens that were CD4+ T cell rich at baseline. Moreover, although the specific B cell lineages affected by depletion was different depending on whether CD4+ or CD8+ T cells were experimentally depleted, the greater effect in CD4+ T cell rich samples was consistent. This data in combination with the viral transcript data shown in Fig 3H–3J reveals that KSHV infection of B cells and the lytic/latent balance is sensitive to the host-specific overall immunological microenvironment and highlight that there is complexity to this relationship that cannot be adequately understood in the context of the current study. It will be interesting in future studies to explore the contribution of donor-specific and context-specific immunology to KSHV lytic reactivation in tonsillar B lymphocyte lineages.

## Materials and methods

### Ethics statement

Human specimens used in this research were de-identified prior to receipt from NDRI and thus were not subject to IRB review as human subjects research.

### Reagents and cell lines

CDw32 L cells (CRL-10680) were obtained from ATCC and were cultured in DMEM supplemented with 20% FBS (Sigma Aldrich) and Penicililin/Streptomycin/L-glutamine (PSG/

Corning). For preparation of feeder cells CDw32 L cells were trypsinized and resuspended in 15 ml of media in a petri dish and irradiated with 45 Gy of X-ray radiation using a Rad-Source (RS200) irradiator. Irradiated cells were then counted and cyropreserved until needed for experiments. Cell-free KSHV.219 virus derived from iSLK cells [39] was a gift from Javier G. Ogembo (City of Hope). Human tonsil specimens were obtained from NDRI. Human fibroblasts were derived from primary human tonsil tissue and immortalized using HPV E6/E7 lentivirus derived from PA317 LXSN 16E6E7 cells (ATCC CRL-2203). Antibodies for flow cytometry were from BD Biosciences and Biolegend and are detailed below.

## Isolation of primary lymphocytes from human tonsils

De-identified human tonsil specimens were obtained after routine tonsillectomy by NDRI and shipped overnight on wet ice in DMEM+PSG. All specimens were received in the laboratory less than 24 hours post-surgery and were kept at 4°C throughout the collection and transportation process. Lymphocytes were extracted by dissection and maceration of the tissue in RPMI media. Lymphocyte-containing media was passed through a 40μm filter and pelleted at 1500rpm for 5 minutes. RBC were lysed for 5 minutes in sterile RBC lysing solution (0.15M ammonium chloride, 10mM potassium bicarbonate, 0.1M EDTA). After dilution to 50ml with PBS, lymphocytes were counted, and pelleted. Aliquots of $5(10)7$ to $1(10)8$ cells were resuspended in 1ml of freezing media containing 90% FBS and 10% DMSO and cryopreserved until needed for experiments.

## Infection of primary lymphocytes with KSHV

Lymphocytes were thawed rapidly at 37°C, diluted dropwise to 5ml with RPMI and pelleted. Pellets were resuspended in 1ml RPMI+20%FBS+100μg/ml DNaseI+ Primocin 100μg/ml and allowed to recover in a low-binding 24 well plate for 2 hours at 37°C, 5% CO2. After recovery, total lymphocytes were counted and Naïve B cells were isolated using Mojosort Naïve B cell isolation beads (Biolegend 480068) or Naïve B cell Isolation Kit II (Miltenyi 130-091-150) according to manufacturer instructions. Bound cells (non-naïve B and other lymphocytes) were retained and kept at 37°C in RPMI+20% FBS+ Primocin 100μg/ml during the initial infection process. $1(10)6$ Isolated naïve B cells were infected with iSLK-derived KSHV.219 (dose equivalent to the ID20 at 3dpi on human fibroblasts) or Mock infected in 400ul of total of virus + serum free RPMI in 12x75mm round bottom tubes via spinoculation at 1000rpm for 30 minutes at 4°C followed by incubation at 37°C for an additional 30 minutes. Following infection, cells were plated on irradiated CDW32 feeder cells in a 48 well plate, reserved bound cell fractions were added back to the infected cell cultures, and FBS and Primocin (Invivogen) were added to final concentrations of 20% and 100μg/ml, respectively. Cultures were incubated at 37°C, 5% CO2 for the duration of the experiment. At 3 days post-infection, cells were harvested for analysis by flow cytometry.

## Flow cytometry staining and analysis of KSHV infected tonsil lymphocytes

A proportion of lymphocyte cultures at baseline or 3dpi representing ~500,000 cells were pelleted at 1400 rpm for 3 minutes into 96-well round bottom plates. Cells were resuspended in 100μl PBS containing (0.4ng/ml) fixable viability stain (BD 564406) and incubated on ice for 15 minutes. Cells were pelleted and resuspended in 100μl cold PBS without calcium and magnesium containing 2% FBS,0.5% BSA (FACS Block) and incubated on ice for 10 minutes after which 100μl cold PBS containing 0.5% BSA and 0.1% Sodium Azide (FACS Wash) was added. Cells were pelleted and resuspended in FACS Wash containing B cell phenotype panel as follows for 15 minutes on ice: (volumes indicated were routinely used for up to $0.5(10)^6$ cells

and were based on titration of the individual antibodies on primary tonsil lymphocyte specimens) CD19-PerCPCy5.5 (2.5μl, BD 561295), CD20-PE-Cy7 (2.5ul, BD 560735), CD38-APC (10μl, BD 555462), IgD-APC-H7 (2.5μl, BD 561305), CD138-v450 (2.5μl, BD 562098), CD27-PE (10μl BD 555441). After incubation, 150μl FACS Wash was added and pelleted lymphocytes were washed with a further 200μl of FACS Wash prior to being resuspended in 200μl FACS Wash for analysis. Data was acquired on a BD FACS VERSE Flow Cytometer and analyzed using FlowJo software. Readers should note that the BD FACS VERSE analysis instrument lacks a 561nm laser so RFP lytic reporter expression from the KSHV.219 genome is not detectable in the PE channel. For baseline T cell frequencies 0.5e6 cells from baseline uninfected total lymphocyte samples were stained and analyzed as above with phenotype antibody panel as follows: CD95-APC (2μl, Biolegend 305611), CCR7-PE (2μl, BD 566742), CD28-PE Cy7 (2μl, Biolegend 302925), CD45RO-FITC (3μl, Biolegend 304204), CD45RA-PerCP Cy5.5 (2μl, 304121), CD4-APC H7 (2μl, BD 560158), CD19-V510 (3μl, BD 562953), CD8-V450 (2.5μl, BD 561426)

## B lymphocyte lineage isolation by cell sorting

At 3 days post-infection cells were collected and pelleted at 1400 rpm for 3 minutes into 12x75mm round bottom tubes. Cells were resuspended in 200μl PBS containing (0.4ng/ml) fixable viability stain (BD 565388) and incubated on ice for 15 minutes. Cells were pelleted and resuspended in 200μl cold MACS buffer containing 5% FBS (Sort Block) and incubated on ice for 10 minutes after which 200μl cold MACS buffer was added. Cells were pelleted and resuspended in MACS buffer containing B cell phenotype panel as follows for 15 minutes on ice: (volumes indicated are for each $1(10)^6$ cells and were scaled depending upon the number of cells being stained for sorting). For single cell sorting of plasma cells the panel was follows: CD19-PerCPCy5.5 (5μl, BD 561295), CD20-PE-Cy7 (5ul, BD 560735), and CD138-APC (5μl, Biolegend 352307). For other lineages the panel was as follows: CD19-PerCPCy5.5 (5μl, BD 561295), CD38-APC (20μl, BD 555462), IgD-PE-Cy7 (5, BD 561314), CD27-PE-Cy5 (5μl eBioscience 15-0279-42). After incubation, 500μl MACS buffer was added and pelleted lymphocytes were washed with a further 500μl of MACS buffer prior to being resuspended in 200μl MACS buffer and put through a cell strainer before sorting using the 70-micron nozzle on a BD FACSAria Fusion Cell Sorter.

## RT-PCR

At 3 days post infection, lymphocytes were harvested into Trizol or sorted into Trizol LS reagent 300μl Trizol was used for >1e5 cells and 100μl Trizol was used for <1e5 cells. An equal volume of DNA/RNA shield (Zymo Research R110-250) was added after collection and RNA extraction was performed using Zymo Directzol Microprep (Zymo Research R2060) according to manufacturer instructions. RNA was eluted in 10μl H2O containing 2U RNase inhibitors and a second DNase step was performed for 30 minutes using the Turbo DNA-Free kit (Invitrogen AM1907M) according to manufacturer instructions. One-step RT-PCR cDNA synthesis and preamplification of GAPDH, LANA and K8.1 transcripts was performed on 5μl of RNA using the Superscript III One-step RT-PCR kit (ThermoFisher 12574026) and 2μM outer primers for each target gene as follows: GAPDH outer forward (5'-TCGGAGTCAACG GATTTGGT-3'), GAPDH outer reverse (5'- GGGTCTTACTCCTTGGAGGC-3'), LANA outer forward (5'-AATGGGAGCCACCGGTAAAG-3'), LANA outer reverse (5'- CGCCC TTAACGAGAGGAAGT-3'), K8.1 outer forward (5'- ACCGTCGGTGTGTAGGGATA-3'), K8.1 outer reverse (5'- TCGTGGAACGCACAGGTAAA-3'). Duplicate no RT (NRT) control reactions were assembled for each lineage/sample containing only Platinum Taq DNA polymerase (Thermofisher 15966005) instead of the Superscript III RT/Taq DNA polymerase mix.

After cDNA synthesis and 40 cycles of target pre-amplification, 0.002µl of pre-amplified cDNA or NRT control reaction was used as template for multiplexed real-time PCR reactions using TaqProbe 5x qPCR MasterMix -Multiplex (ABM MasterMix-5PM), 5% DMSO, primers at 900nM and probes at 250nM against target genes as follows: GAPDH forward (5'-TCGG AGTCAACGGATTTGGT-3'), GAPDH reverse (5'- GGGTCTTACTCCTTGGAGGC-3'), GAPDH probe (5'[HEX]-ACGCCACAGTTTCCCGGAGG-[BHQ1]3') LANA forward (5'-A ATGGGAGCCACCGGTAAAG-3'), LANA reverse (5'- CGCCCTTAACGAGAGGAAGT-3'), LANA probe (5' [6FAM]-ACACAAATGCTGGCAGCCCG-[BHQ1]3'), K8.1 forward (5'- AC CGTCGGTGTGTAGGGATA-3'), K8.1 reverse (5'- TCGTGGAACGCACAGGTAAA-3'), K8.1 probe (5'[FAM]-TGCGCGTCTCTTCCTCTAGTCGTTG-[TAMRA]3') and analyzed using a 40 cycle program on a ThermoFisher Quantstudio 3 real time thermocycler. Data is represented as quantitation cycle (Cq) and assays in which there was no detectable Cq value were set numerically as Cq = 41 for analysis and data visualization.

## Single cell RT-PCR

At 3 days post-infection, cells were stained for sorting as described above. Single cells meeting lineage sort criteria were sorted into each well of a 0.2ml 96-well PCR plate containing 4µl of 0.5x PBS, 10mM DTT (Pierce no-weigh A39255), 1.2U RNase inhibitor (Lucigen 30281–2). After sorting, plates were sealed, centrifuged briefly to collect all material in the bottom of the well and stored at -80˚C prior to analysis. Plates were thawed on ice and 2µl of DNase buffer (Invitrogen 18068–015) containing 0.5µl 10x buffer, 0.1U DNase, 0.4µl H2O) were added to each well. After incubation at room temperature for 15 minutes, EDTA (Thermofisher AM9260G) was added to a final concentration of 2mM and DNase was inactivated by incubation for 10 minutes at 65˚C. One-Step RT-PCR reactions and no RT (NRT) controls were assembled using outer primers to GAPDH, LANA, and K8.1 as described above. 2µl of pre-amplified cDNA was used in the real time PCR reactions for GAPDH, K8.1, LANA as described above with the exception that the assay was multiplexed with all three targets using the same K8.1 probe sequence labeled with 5'Cy5 and 3'BHQ-2 quencher and analyzed on a Biorad CFX96 real time PCR thermocycler.

## KSHV neutralization via soluble CD138 Syndecan-1

Infections were performed as described above except KSHV.219 virus was pre-incubated for 30 minutes on ice with serum free RPMI only or serum free RPMI containing recombinant human syndecan-1 protein (srCD138, BioVision, 7879–10) prior to being added to Naïve B lymphocytes. srCD138 concentrations noted in the text indicate the final concentration of recombinant syndecan-1 in the reconstituted total lymphocyte culture. Infection was analyzed at 3 days post-infection by flow cytometry for B cell lineages and KSHV infection as detailed above. For experiments involving human fibroblasts virus was added to cells in serum free media, cells were spinoculated for 30 minutes at 1000rpm, incubated at 37˚C for 1 hour, then infection media was removed and replaced with complete media. At 3dpi cells were harvested via trypsinization and analyzed for infection by flow cytometry.

## Heparinase treatment of human fibroblasts and lymphocytes

Total lymphocytes were thawed and recovered as described above and treated with Heparinase I and III Blend from Flavobacterium heparinum (Sigma-Aldrich # H3917) at 9U/25e6 of lymphocytes and incubated over irradiated CDW32 feeder cells at 37˚C, 5% CO2 for 24 hours in complete media. E6/E7 transformed fibroblasts from human tonsil were treated at 4.5U/1e6 for 24hrs at 37˚C, 5% CO2. To evaluate the effectiveness of the Heparinase treatment the

samples were stained for flow cytometry, as described above, with Heparan Sulfate (10E4 epitope) (FITC) (United States Biological # H1890) at 2 μl/5e5 of the lymphocytes or E6/E7 transformed fibroblasts and analyzed with flow cytometry for loss of HSPG signal. Control untreated or Heparinase-treated samples were either Mock infected or infected with KSHV as described above. At 3 days post infection, fibroblasts were trypsinized and analyzed for GFP expression by flow cytometry and lymphocytes were harvested, stained for B cell lineages and analyzed by flow cytometry as described above.

### T cell depletion studies

Infections were performed as described above except a sub-population of total lymphocytes were depleted of either CD4 or CD8 T cells using positive selection magnetic beads (Biolegend MojoSort Human CD4 T Cell Isolation Kit 480009, MojoSort Human CD8 T Cell Isolation Kit 480011). The resulting depleted fractions or unmanipulated total lymphocytes were used to reconstitute naïve B lymphocytes following infection rather than bound lymphocyte fractions as described above.

Statistical Analysis. Data plots and statistical analysis were performed in R software[40] using ggplot2[41] ggcorrplot{ggcorrplot:2018tg} and RColorBrewer[42] packages. Specific methods of statistical analysis and resulting values for significance and correlation are detailed in the corresponding figure legends.

## Supporting information

**S1 Fig. Gating schemes for tonsil lymphocyte lineages.** Flow cytometry data for a baseline uninfected sample from a 2-year-old male donor showing representative gating and lineage definitions used in the study for (A) B cell lineages based on vanZelm et. al. 2007 [43] and (B) T cell lineages based on Mahnke et. al. 2013 [44].
(TIF)

**S1 Table. Analyzed flow cytometry data.** Values derived from flow cytometry analysis for baseline B cell and T cell lineage frequencies, overall infection frequency at 3dpi and lineage-specific infection frequencies for B cells. Comments associated with column headers contain detailed definitions for each subset.
(XLSX)

**S2 Table. Analyzed flow cytometry data for B cell lineages at 3 days post infection.** Values derived from flow cytometry analysis for overall B cell lineage frequencies in Mock and KSHV-infected cultures at 3 days post-infection. Abbreviations and lineage definitions are as in S1 Table comments.
(XLSX)

## Acknowledgments

The authors thank Dr. J. Gordon Ogembo for providing the KSHV.219 virus that was used in this study, Dr. Charles R. Rinaldo for valuable feedback on the preprint version of this manuscript, and Emily Romano for critical reading of the revised manuscript.

## Author Contributions

**Conceptualization:** Jennifer Totonchy.

**Data curation:** Jennifer Totonchy.

**Formal analysis:** Romina Nabiee, Jesus Ramirez Castano, Jennifer Totonchy.

**Funding acquisition:** Jennifer Totonchy.

**Investigation:** Farizeh Aalam, Romina Nabiee, Jesus Ramirez Castano, Jennifer Totonchy.

**Methodology:** Farizeh Aalam, Jennifer Totonchy.

**Project administration:** Jennifer Totonchy.

**Resources:** Jennifer Totonchy.

**Supervision:** Jennifer Totonchy.

**Validation:** Farizeh Aalam, Romina Nabiee, Jennifer Totonchy.

**Visualization:** Jennifer Totonchy.

**Writing – original draft:** Farizeh Aalam, Jennifer Totonchy.

**Writing – review & editing:** Farizeh Aalam, Romina Nabiee, Jesus Ramirez Castano, Jennifer Totonchy.

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
