## [Decision Letter · Decision Letter 0]

27 Apr 2020

Dear Dr. Totonchy,

Thank you very much for submitting your manuscript "Analysis of KSHV B lymphocyte lineage tropism in human tonsil reveals efficient infection of CD138+ plasma cells" for consideration at PLOS Pathogens. As with all papers reviewed by the journal, your manuscript was reviewed by members of the editorial board and by several independent reviewers. In light of the reviews (below this email), we would like to invite the resubmission of a significantly-revised version that takes into account the reviewers' comments.

Overall the study was deemed of significance, particularly due to the potential relevance of direct infection of CD138 positive B cells for viral pathogenesis and the use of primary lymphoid organ cell cultures.

However, the study was also considered descriptive by several reviewers, with minimal mechanistic insight and some doubt over the degree of contribution to the field over the already published work of the authors and other groups.

While responses to all reviewers' points are expected, the particular emphasis should be placed on defining the viral gene expression and type of infection, as requested by several reviewers, providing some mechanistic insight into viral preference for select B cell subsets and the phenotypes observed with T cell depletion, and addressing important technical issues brought up in the major concerns section of reviewer #2.

We cannot make any decision about publication until we have seen the revised manuscript and your response to the reviewers' comments. Your revised manuscript is also likely to be sent to reviewers for further evaluation.

Sincerely,

Vera L Tarakanova

Guest Editor

PLOS Pathogens

Shou-Jiang Gao

Section Editor

PLOS Pathogens

Kasturi Haldar

Editor-in-Chief

PLOS Pathogens

orcid.org/0000-0001-5065-158X

Michael Malim

Editor-in-Chief

PLOS Pathogens

orcid.org/0000-0002-7699-2064

Overall the study was deemed of significance, particularly due to the potential relevance of direct infection of CD138 positive B cells for viral pathogenesis and the use of primary lymphoid organ cell cultures.

However, the study was also considered descriptive by several reviewers, with minimal mechanistic insight and some doubt over the degree of contribution to the field over the already published work of the authors and other groups.

While responses to all reviewers' points are expected, the particular emphasis should be placed on defining the viral gene expression and type of infection, as requested by several reviewers, providing some mechanistic insight into viral preference for select B cell subsets and the phenotypes observed with T cell depletion, and addressing important technical issues brought up in the major concerns section of reviewer #2.

Reviewer's Responses to Questions

**Part I - Summary**

Reviewer #1: The authors have analyzed KSHV infection of 40 human tonsils, focusing on infection of tonsillar B cells. Their approach was methodical (though did involve freezing and thawing the lymphocytes prior to infection) and employed a feeder cell system to maintain lymphocyte viability that they had published previously. Their main finding was that KSHV appears to have a strong predilection for CD138+ cells, which is of interest since one of the main diseases from KSHV (MCD) are exclusively CD138+. The strength of this work is the careful flow-cytometry based analysis of multiple B cell subtypes in coming to this conclusion. The other cell types really did not compare in terms of susceptibility to infection.

Their approach (negative selection of B cells) involved the post-infection addition of cells originally selected against. This is artificial since it is a far cry from the natural T/B/connective tissue environment of the tonsil's microenvironment but, nevertheless, did enable their detailed B cell subtype analyses.

In addition, the investigators examined the role of CD138 as an actual viral receptor, critical to the tropism that they noted. Their results were inconclusive and they concluded that other factors likely play a role.

One potential weakness in their overall system is their dependence on a red/green reporter system and a single (3 dpi) timepoint that would allow GFP signal persistence. See "Major issues" below. The authors do not, at least in this study on B cell subtypes, verify that their GFP expression represents true ("stable") infection with viral gene expression. The KSHV.219 GFP is a marvelous surrogate marker for entry but reflects transcription of a cellular promoter. Further, the authors criticized other studies in the Discussion for looking at only latent or lytic gene expresso but then only analyzed GFP, which represents neither. (The authors could have examined their cells for lytic activity easily by looking for red fluorescence and may have but didn't report it .) Finally, if lytic activity did occur, CD138 by 3dpi could be down-regulated below flow cytometric detection infection, as others have shown with reactivation systems. Though unlikely, such down-regulation could possibly skew their conclusion.

Reviewer #2: Transmission of KSHV is presumed to be via saliva, but it is not clear what cells in the nasopharyngeal cavity are infected. This is a comprehensive analysis to determine which B cell subsets are permissive for KSHV infection in the tonsillar tissue of 40 donors of various ages. The authors utilize a GFP reporter virus coupled with high resolution multi-parameter flow cytometry to monitor cells that are infected over a 3 day co-culture experiment. Even though there is a wide range of GFP+ infection donors, several trends manifest. The major finding is that multiple subsets of B cells at different stages of differentiation are targets for KSHV- but of these, CD138+ plasma cells are infected at the highest frequency. Tropism was not accounted for merely based on syndecan/CD138 on the surface of plasma cells since soluble CD138 pre incubation with virions did not abrogate infection of the CD138+ cells. They also observed that a depletion of CD4+ T cells in donor samples with a higher baseline ratio of CD4 to CD8 cells led to a slight increase in GFP+ B cells. This paper further expands our understanding of KSHV tropism in primary B lymphocytes. However, there is no analysis of viral gene expression, cytokine profiles, or examination of T cell effector responses that might provide mechanistic insight. Critical technical controls are lacking to confirm that the virus adsorption with soluble CD138 and T cell ablation strategies were effective.

Reviewer #3: KSHV is a B cell tropic and lymphomagenic virus transmitted via saliva. Therefore, defining the specific B-cell tropism in the context of the tonsillar environment is essential to fully understand KSHV transmission and the pathobiology of de novo infection in the B-cell lineage with essential implications for KSHV induced lymphomagenesis. In this succinct and well-designed study, the Totonchy group uses a collection of 40 human tonsils to identify the B-cell lineage, which is most susceptible to KSHV infection. They found that CD138+ plasma cells, albeit constituting a small percentage of the B-cell lineage population are highly infectable by KSHV. They also provide data pointing to a role of CD138 in virus attachment and that T-cells present in the microenvironment may influence the outcome of infection. Thus, this is an important and significant study; mainly, since one of the markers of the PEL KSHV driven lymphomas is CD138, the study suggests that these lymphomas may arise upon infection of more mature and differentiated B-cells. It opens new avenues for the development of models of KSHV lymphomagenesis. A few experiments will make it even more robust and compelling. Particularly important would be characterizing the type of infection carried out by KSHV in the described population.

**Part II – Major Issues: Key Experiments Required for Acceptance**

Reviewer #1: 1. Infection determination: KSHV.219 expresses GFP off of the constitutively acvtive cellular (i.e. non-viral) EF-1α promoter. Thus, detection of GFP indicates only that the viral genome entered the cell nucleus but not that it established bona fide infection. To ascertain biologically relevant levels of infection (rather than simply entry, which is necessary but not sufficient for the establishment of infection), the authors would need to assess either latent and/or lytic native gene expression. For the latter, KSHV.219 would allow RFP to serve as a surrogate since it’s driven by the KSHV lytic PAN promoter. This point is important since it is absolutely conceivable that various tonsillar lymphocytes could be permissive to entry but not support downstream viral gene expression (and establishment of actual infection, whether latent or lytic).

2. Fig. 4: If the CD138 is going to be tested as a contributor to entry in this system, it would have been helpful to have a negative control, especially since no dose-response of the srCD138 is apparent. E.g. incubating with another recombinant surface protein; looking at the results from blocking CD138 on the surface of cells prior to addition of the virus; or showing that another virus not known to use CD138 is not blocked by the addition of srCD138. Overall, the authors conclude that the data are not particularly convincing so perhaps this fig. could be omitted altogether.

Reviewer #2: 1. Technical concerns with the infection experiments.

• A tonsillectomy is presumed to be in response to a medical issue. Were there any donors with specific conditions that led to disparate, skewed results as outliers? Were co-infections with adenovirus or EBV examined?

• These cells were frozen prior to use and plasma cells have decreased recovery after freezing. Was the baseline flow analysis performed at the initial isolation prior to freezeback or subsequent to thawing after/concurrent with infection?

• As described for Figure 2A procedure, 1x10^6 naïve B cells were used for infection to normalize infection across donors. The authors should clarify whether uninfected naïve cells were added back to the co-culture to restore naïve B cell numbers or was there a constant ratio of infected naïve to bound cells across all donors?

• Are the naïve cells truly infected or are they just serving as carriers of cell-surface bound virions for the 3 day culture?

• Do the bound cells have some baseline level of activation after the naïve B cell isolation? After incubation on the feeder cells?

• The shCD138 studies are not conclusive. How much recombinant CD138 was added? There was no verification that the CD138 bound to the virions. Can the CD138 on the targets cells be blocked?

• Figure 5. The increase in GFP+ cells is only increased by 20% and that is influenced heavily by 3 of the donors. Is this increase statistically significant in a paired T test for the CD4 depletion?

• Line 149: Why is T cell depletion data not shown? This is a critical control that should be included, minimally as a supplemental figure.

2. Data analysis and presentation of the data

• Figure 1C- Is there a direct correlation between increased memory and loss of germinal center B cells in donors by age?

• Figure 2B is very difficult to interpret. The authors might place GFP on the y axis and break down the data as in Figure 1A by gender and race with a color code and then have a separate panel with GFP on y axis with the donor age bins on the x-axis as presented in Fig 1C. What statistical test was applied to determine a lack of correlation with age, gender, or race in Figure 2?

• Figure 3 highlights the CD138 as having the highest percentage of infection, but this does definitively demonstrate that CD138+ cells are the major latency reservoir. An analysis based on total numbers in the tonsil tissue (% infected * frequency of total) provided as a table or additional graph is needed to describe how the infected cell subsets contribute to the overall viral load in the tonsil tissue.

• This group previously reported an Ig lambda light chain bias in the KSHV+ cells and an increase in IL-6 production in the infected cultures. There are likely other features of B cell maturation, differentiation and survival that are influenced by infection. Did the authors observe any bias in Ig light chains of the plasma cells in this study?

• Figure 3C examines how the percent GFP in a given subset correlates with baseline frequency, but a similar analysis with GFP correlation to 3 dpi would also be helpful and complement Fig 3A and 3B. Does the negative 0.4 correlation of plasma cells with memory in Fig 3C indicate the virus is shifting memory to plasma in the culture or that if there are more memory at baseline there will just be fewer plasma cells to infect? Does the subset distribution (e.g. CD27) change between 0 and 3 dpi with KSHV infection? Do marginal zone or plasma cells increase?

• Figure 4. The % changes in GFP are variable by donor. Is this attributed to a very low number of cells being infected such that any small changes can lead to positive or negative changes in the % infected which are not biologically meaningful? Were the 0 dose samples infected with a mock virion stock pre-incubated with the rhCD138 vehicle? Could the individual donors be color-coded across the subsets of the figure?

• The T cell composition of supplemental figure 2 was not well-incorporated into the paper. Did any particular T cell subsets correlate with % infected B cells?

• With regard to mechanism, were viral genes profiled to examine differences in the major subsets (naïve, marginal zone, germinal center, memory plasma cells? Did any cytokine profiles correlate with B cell subsets, T cell subsets (before/after depletion), or percent infections?

3. Interpretations and incorporation of findings into larger field

• The results section is quite brief and does not fully explain the data or develop a meaningful narrative.

• What stage of virus infection does the GFP reporter gene indicate? If an artificial promoter, could the signal strength differ with cell type and activation status?

• Please expand on the discussion comment that these observations complement the finding of Myoung et al. showing that CD4+ T cells control lytic reactivation of KSHV in primary B cells.

• The reader needs more information on the UPR and KSHV discussion point; it is rather vague without referring to the cited paper.

• How does this subset distribution relate the EBV in cell culture or MHV68 in vivo?

• How would CD20+ plasma vs CD20- plasma vs plasmablasts contribute to infection/transmission and pathogenesis/disease?

Reviewer #3: 1. An essential aspect of completing this study is the characterization of the type of KSHV infection established in the CD138+ Plasma cells. I think that since the researchers are using a r219KSHV, it is a missed opportunity that they do not try to procure a proper cytofluorometer to take advantage of this opportunity. If that continues not to be affordable, they can at least use qRT-PCR of a panel of KSHV RTA and early lytic, late lytic and latent genes and by test virion production by testing the infectivity of the supernatants in 293 cells.

2. This group has previously shown the ability of KSHV infection to manipulate the Ig recombination machinery in naïve B cells and other groups had shown the potential of KSHV infection to induce Ig heavy chain class switch recombination. Therefore it will be interesting to determine the existence of changes in the Ig class population composition upon infection of CD138+ plasma cells.

3. The results with the srCD138 showing the potential involvement of this receptor in KSHV attachment are interesting. It is worthy to notice that since this mechanism is mediated via HSPGs the researchers may not obtain the kind of clear-cut results obtained when testing a major internalizing receptor since HSPGs generally contribute to binding to the cell surface so that the specific receptor can internalize the virus. It will be interesting to test whether heparin treatment of the virus or heparitinase/ heparanase pre-treatment of the target cells mimics the results obtained with the soluble CD138.

**Part III – Minor Issues: Editorial and Data Presentation Modifications**

Reviewer #1: Fig. S1A. B and T cells based on 31 and 32? To what do 31 and 32 refer?

‘fig. 2 B. It is unclear in the text and legend what each filled-in circle represents. If it indicates a separate tonsil, then why are there >40 circles? Suggest clarification in legend.

Line 97: susceptibility of “naïve” (insert “naïve”) B cells to KSHV…

Lines 118-121

Conclusion (1) and (2) is consistent with their data but also consistent with findings from previous work from other labs. Consider inserting such a statement.

Line 189. Not all studies used stimulation of target cells in in vitro infection systems.

Mx discussion might be improved if authors integrated their data with earlier reports of tonsilar KSHV+ cells demonstrating plasmablast-like phenotype.

Reviewer #2: • Line 45: ‘unstimlated’

• Referred to as srCD138 in text, but rhCD138 in figure 4.

• The frequent use of the singular ‘tonsil’ seems awkward.

• Figure legends are a bit sparse and do not enable the figures to stand alone.

• Figure 5 y axis legend labels: specify percent of non-depleted control

• Figure 5A: specify what the colors indicate.

• No label 7 in supp figure 1

• Hard to distinguish labels with specific boxes in supplemental figure

Reviewer #3: Since this is a succinct study, it will help the reader that the supplementary data are presented in the main text figures. These include patient data in classic table form, B-cell lineages etc.

PLOS authors have the option to publish the peer review history of their article (what does this mean?). If published, this will include your full peer review and any attached files.

Reviewer #1: No

Reviewer #2: No

Reviewer #3: No
---

## [Editor Report · Decision Letter 1]

7 Sep 2020

Dear Dr Totonchy,

We are pleased to inform you that your manuscript 'Analysis of KSHV B lymphocyte lineage tropism in human tonsil reveals efficient infection of CD138+ plasma cells' has been provisionally accepted for publication in PLOS Pathogens.

Best regards,

Vera L Tarakanova

Guest Editor

PLOS Pathogens

Shou-Jiang Gao

Section Editor

PLOS Pathogens

Kasturi Haldar

Editor-in-Chief

PLOS Pathogens

orcid.org/0000-0001-5065-158X

Michael Malim

Editor-in-Chief

PLOS Pathogens

orcid.org/0000-0002-7699-2064
---

## [Editor Report · Acceptance letter]

9 Oct 2020

Dear Dr Totonchy,

We are delighted to inform you that your manuscript, "Analysis of KSHV B lymphocyte lineage tropism in human tonsil reveals efficient infection of CD138+ plasma cells," has been formally accepted for publication in PLOS Pathogens.

Best regards,

Kasturi Haldar

Editor-in-Chief

PLOS Pathogens

orcid.org/0000-0001-5065-158X

Michael Malim

Editor-in-Chief

PLOS Pathogens

orcid.org/0000-0002-7699-2064